# Global, regional, and national burden of near vision loss in children and adolescents under 20 years from 1990–2021 and prediction to 2060: A cross-sectional study based on the global burden of disease study 2021

Jing Peng[1,2], Cong Zhang[1,2], Xingying Li[1,2], Jin Tang[3], Xi-Yuan Zhou[1,2]*

1 Department of Ophthalmology, The Second Affiliated Hospital of Chongqing Medical University, Chongqing, China, 2 Key Laboratory of Ophthalmology of Chongqing Municipality, Chongqing, China, 3 Department of Orthopaedic Surgery, The Second Affiliated Hospital of Chongqing Medical University, Chongqing, China

* zhouxiyuan2002@aliyun.com

## Abstract

Near vision loss (NVL) has become a significant global public health concern, particularly among children and adolescents under 20 years, who face increasing academic demands and prolonged screen exposure. The COVID-19 pandemic, characterized by excessive screen time and reduced outdoor activities, has likely exacerbated this trend. This study analyzes the global, regional, and national burden of NVL from 1990 to 2021 and projects future trends up to 2060 using data from the Global Burden of Disease (GBD) Study 2021. Prevalence and Disability-Adjusted Life Years (DALYs) associated with NVL were assessed across different socioeconomic levels, and future trends were forecasted using the Bayesian Age-Period-Cohort (BAPC) model. Results indicate a significant increase in NVL cases, rising to 31.7 million in 2021, with projections reaching 33 million by 2060. A strong negative correlation was observed between the Social Development Index (SDI) and NVL burden, with Africa exhibiting the highest prevalence and Australasia the lowest. Notably, NVL burden in higher SDI regions rebounded post-COVID-19, reversing previous declining trends. Across all age groups, NVL prevalence continues to rise, with females consistently exhibiting higher rates than males. These findings underscore the urgent need for targeted public health policies and resource allocation strategies to mitigate the rising burden of NVL among children and adolescents, particularly in lower SDI regions. Addressing modifiable risk factors, promoting early interventions, and integrating vision care into public health frameworks will be crucial in managing this growing health crisis.

**Data availability statement:** All relevant data are within the paper and its Supporting information files.

**Funding:** The author(s) received no specific funding for this work.

**Competing interests:** The authors have declared that no competing interests exist.

## Introduction

Near vision loss (NVL) is a leading cause of visual impairment and blindness, posing significant challenges to global health systems [1,2]. NVL is driven by factors such as reduced outdoor time, increased near-work activities, and rising urbanization in developing countries [3]. Past research evidence suggests that the prevalence of NVL is increasing worldwide and is expected to reach 50% globally by 2050 in the absence of effective interventions [4]. Traditionally, NVL has been predominantly associated with aging populations [5]. For example, older people have more ocular symptoms, such as cataract, glaucoma, retinal detachment, and myopic macular degeneration, all of which can cause irreversible vision loss [6]. This will pose a huge challenge to disease management and put enormous pressure on global health-care systems [7]. One study estimated that in 2015 there was about $244 billion in potential global productivity losses associated with the burden of NVL [7]. However, recent trends indicate a growing prevalence among children and adolescents under 20 years, driven by increasing risk factors such as prolonged near-work activities, reduced outdoor exposure, and widespread use of digital devices. These factors are particularly prevalent in modern societies, where academic demands and sedentary lifestyles contribute to extended periods of near-vision strain [8,9].

The COVID-19 pandemic has further exacerbated the situation. Lockdowns and social distancing measures led to a dramatic increase in screen time and a reduction in outdoor activities [9]. Both of which are known risk factors for NVL [10]. Studies have shown a significant acceleration in the prevalence of NVL among children and adolescents during this period, highlighting the urgent need for targeted interventions [11–13].

Despite these emerging trends, the global burden of NVL after COVID-19 in children and adolescents remains underexplored. Previous analyses have documented a decline in overall blindness and vision loss among this demographic between 1990 and 2019 [14]. However, the impact of the COVID-19 pandemic has reversed this trend, with significant increases in NVL prevalence reported in many regions [11–13,15]. Therefore, identifying temporal, geographic, and locational trends in the burden of NVL are essential for developing effective prevention strategies and allocating treatment resources.

The Global Burden of Disease (GBD) study provides a comprehensive framework for assessing the impact of NVL across different regions and time periods [16–18]. The GBD 2021 study, in particular, offers critical insights into the changes in disease burden following the COVID-19 outbreak. Therefore, this study aims to provide a detailed assessment of the global, regional, and national burden of NVL among children and adolescents under 20 years from 1990 to 2021, using the latest data from the GBD 2021 database. Additionally, we project the future burden of NVL up to 2060 to inform timely adjustments in diagnosis, treatment, and prevention strategies.

## Methods

### Data source and disease definition

This study adheres strictly to the STROCSS reporting criteria [19].
All data were obtained from the Global Health Data Exchange platform

(http://ghdx.healthdata.org/gbd-results-tool). Prevalence and DALYs data for NVL were extracted from the 2021 GBD Study [17]. The 2021 GBD study provided the foundational data into the burden of NVL, including prevalence, DALYs, and their corresponding rates and uncertainty intervals across 369 diseases and injuries in 204 countries and territories from 1990 to 2021 [16,17]. NVL in the GBD 2021 Study is defined as *presenting near vision worse than N6 or N8 at 40 cm, with best-corrected distance visual acuity preserved at 6/12 or better*. While this metric primarily identifies uncorrected presbyopia in adults, pediatric NVL may also result from hyperopia, astigmatism, or amblyopia. Cases with concurrent distance vision loss (acuity <6/12) were excluded to isolate near-vision-specific impairment.

### Age stratification rationale

The upper age limit of 20 years aligns with the GBD's pediatric categorization, which emphasizes developmental stages and intervention efficacy. Early vision care (e.g., refractive correction, amblyopia therapy) is most impactful before visual plasticity declines in late adolescence. Additionally, the 15–19 age group faces heightened risks of screen-induced myopia progression, particularly during the COVID-19 pandemic.

This study collected data on prevalence and DALYs in NVL in children and adolescents under the age of 20 years, along with their corresponding rates at the global, regional, and national levels. The GBD database does not include data on participants' race or ethnicity, as these variables were not assigned for data collection. This study utilized secondary data from this collaborative project without direct participant interaction. No children and adolescents under 20 years of age were involved in formulating research questions, outcome measures, or in the study's design or implementation.

### SDI analysis

The GBD database encompasses 21 regions and 204 countries, categorized by quintiles of the SDI. Detailed information regarding the regional division of the SDI can be accessed through the Institute for Health Metrics and Evaluation at: https://ghdx.healthdata.org/search/site/SDI. The SDI serves as a composite measure that includes per capita income, total fertility rate, and average educational attainment, thereby reflecting a country's overall social and economic development [20]. The relationship between SDI and the burden of NVL was examined by calculating SDI-specific disease rates. SDI categories—low, low-middle, middle, high-middle, and high—were utilized to compare disease burdens across varying levels of socioeconomic development. The "dplyr" and "ggplot2" packages in R were employed for data manipulation and visualization.

### Global and regional burden analysis

To analyze the global distribution and regional differences in the burden of NVL in children and adolescents under the age of 20 years, we created global maps and conducted regional comparative analyses. Data were aggregated by geographical regions as defined in the GBD study. Maps were generated using R (version 4.4.1) with the "ggplot2" and "sf" packages to visualize the spatial distribution of disease burden.

### Prediction of NVL to 2060

To predict the future burden of NVL in adolescent under 20 years of age, we employed the Bayesian Age-Period-Cohort (BAPC) model, implemented in R with the "INLA" and "BAPC" packages. This advanced model forecasts DALYs and prevalence trends up to 2060, integrating age, period, and cohort effects to provide a nuanced understanding of disease progression [21]. By analyzing these dimensions, the BAPC model offers critical insights into the potential evolution of NVL, influenced by demographic shifts, historical health trajectories, and anticipated changes in risk factors and medical interventions.

## Statistical analysis

All statistical analyses and data visualizations were conducted using R (version 4.4.1). We began by extracting NVL prevalence data across multiple years from the GBD database, subsequently cleaning and organizing the data in R to ensure integrity and consistency. Data were categorized by region and age group and formatted for detailed analysis. Descriptive statistics were computed for all key variables, with results presented as means and 95% uncertainty intervals (UIs). For trend analyses, statistical significance was defined as $p < 0.05$. For the all-age population analysis, age-standardized rates (ASRs) per 100,000 individuals were calculated, including the age-standardized prevalence rate (ASPR) and age-standardized DALY rate. The ASR was calculated by the following formula: $ASR = \frac{\sum_{i=1}^{A} a_i w_i}{\sum_{i=1}^{A} w_i} \times 100,000$.

Prevalence and DALYs are reported as age-standardized rates per 100,000 population, consistent with GBD reporting standards. This metric optimizes cross-regional and temporal comparability, particularly for conditions with low absolute case counts.

The Estimated Annual Percentage Change (EAPC) was calculated using linear regression models, with the "broom" package used to process and tidy the regression output. To ensure accurate analysis, the "broom" and "dplyr" packages were used, as they were crucial for accurate analysis, efficiently extracting and organizing regression results, enabling precise calculation and interpretation of the EAPC.

## Ethical approval and consent to participate

Our research involved a secondary evaluation of the publicly accessible GBD Study, without primary data collection. The confidentiality of the participants was ensured during data collection, and informed consent was taken from the respondents during the NFHS survey.

## Results

### Global trends and variations across SDI regions

Globally, the prevalence of NVL among adolescents experienced a substantial increase, rising from approximately 25.3 million cases in 1990 to 31.7 million in 2021, marking a 25.3% surge (Table 1). Over the same period, the DALYs associated with NVL in adolescents exhibited a similar upward trajectory, increasing by 25.4% (Table 2). Both prevalence and DALY rates showed consistent growth trends, with estimated annual percentage changes (EAPCs) of 0.136 and 0.139, respectively (Tables 1 and 2).

Across the five SDI regions, the burden of NVL in individuals under 20 years old exhibited significant disparities in trends. In High SDI, High-middle SDI and Middle SDI regions, ASPR and ASR of DALYs declines after reaching a certain peak, The High and High-middle SDI regions peaked between 1995 and 2005, and the Middle SDI region peaked around 2005, while in Low-middle SDI and Low SDI regions, ASPR shows a yearly increase. Notably, an unfavorable reversal of the previously declining ASPR trend was observed in higher SDI regions after 2020 (Fig 1).

In 2021, low SDI regions bore the highest disease burden, with the highest prevalence and DALY rates recorded at 1,419.199 and 14.765 per 100,000 population, respectively (Tables 1 and 2). These regions also exhibited the highest EAPCs for these indicators, at 0.334 and 0.349, suggesting a sustained and relatively high growth rate (Tables 1 and 2). Conversely, high SDI regions reported the lowest prevalence and DALY rates, at 685.662 and 7.153 per 100,000 population, respectively (Tables 1 and 2), indicating the least disease burden (Table 3).

Of particular concern, low-middle SDI regions demonstrated the most rapid increase in disease burden. Between 1990 and 2021, the change in prevalence and DALYs in these regions rose by 14.045% and 14.343%, respectively (Tables 1 and 2). Moreover, these regions recorded the highest EAPCs for ASPR and ASR of DALYs, at 0.502 and 0.513, highlighting an accelerating growth trend that warrants close monitoring (Tables 1 and 2).

**Table 1. Prevalence of near vision loss between 1990 and 2021 at the Global and Regional Level.**

| Location | 1990 Prevalence cases (95%UI) | Prevalence rate(95%UI) | 2021 Prevalence cases (95%UI) | Prevalence rate(95%UI) | 1990-2021 Rate change (95%UI) | EAPC (95%UI) |
|---|---|---|---|---|---|---|
| Global | 25303382.725 (12540477.515, 45055847.538) | 1120.325 (555.239, 1994.880) | 31737503.807 (15630327.501, 56451013.207) | 1204.075 (592.992, 2141.670) | 7.475 (5.266, 9.917) | 0.136 (0.081, 0.191) |
| **SDI** | | | | | | |
| High-middle SDI | 4323321.015 (2135957.557, 7903204.388) | 1167.942 (577.027, 2135.044) | 3258437.129 (1552928.885, 5870044.501) | 1074.107 (511.905, 1934.994) | −8.034 (−13.845, −3.927) | −0.621 (−0.815, −0.426) |
| High SDI | 1739897.006 (805939.806, 3085387.571) | 692.304 (320.683, 1227.673) | 1595690.799 (738402.087, 2873256.024) | 685.662 (317.288, 1234.626) | −0.959 (−3.456, 1.618) | −0.176 (−0.257, −0.095) |
| Low-middle SDI | 5936138.932 (2874597.955, 10495593.918) | 1004.382 (486.376, 1775.833) | 8755664.508 (4269993.527, 15495344.712) | 1145.448 (558.616, 2027.158) | 14.045 (10.100, 18.268) | 0.503 (0.460, 0.546) |
| Low SDI | 3571809.300 (1803889.605, 6337430.470) | 1277.579 (645.223, 2266.798) | 8291042.606 (4189283.928, 14830523.585) | 1419.199 (717.090, 2538.578) | 11.085 (8.316, 13.688) | 0.334 (0.304, 0.364) |
| Middle SDI | 9712582.047 (4851708.618, 17377602.066) | 1270.360 (634.580, 2272.908) | 9814506.125 (4813942.036, 17459240.357) | 1310.014 (642.552, 2330.412) | 3.121 (0.579, 6.205) | −0.026 (−0.146, 0.094) |
| **Regions** | | | | | | |
| Andean Latin America | 251384.753 (123621.078, 450536.508) | 1326.142 (652.144, 2376.737) | 316751.723 (154771.463, 575043.813) | 1337.998 (653.773, 2429.054) | 0.894 (−7.171, 9.324) | 0.147 (0.107, 0.186) |
| Australasia | 39424.302 (16911.435, 72124.058) | 628.460 (269.584, 1149.725) | 45880.655 (19980.877, 82761.015) | 608.353 (264.936, 1097.367) | −3.199 (−11.014, 5.819) | −0.069 (−0.125, −0.013) |
| Caribbean | 216765.483 (111848.648, 371912.844) | 1435.629 (740.769, 2463.165) | 220500.834 (111387.111, 387069.524) | 1444.717 (729.806, 2536.072) | 0.633 (−3.450, 6.241) | 0.046 (−0.021, 0.113) |
| Central Asia | 280248.055 (130466.885, 499613.593) | 887.426 (413.133, 1582.063) | 307349.841 (144098.984, 557181.444) | 887.662 (416.175, 1609.205) | 0.027 (−3.050, 3.051) | −0.154 (−0.363, 0.057) |
| Central Europe | 249811.490 (115145.113, 466317.820) | 636.165 (293.226, 1187.515) | 147125.724 (66553.977, 273307.123) | 624.546 (282.521, 1160.184) | −1.826 (−4.752, 0.468) | −0.264 (−0.368, −0.161) |
| Central Latin America | 1204212.290 (613632.068, 2091524.629) | 1457.332 (742.615, 2531.154) | 1301514.894 (641937.762, 2357650.686) | 1526.077 (752.697, 2764.437) | 4.717 (−4.443, 10.485) | 0.112 (0.072, 0.151) |
| Central Sub-Saharan Africa | 741566.268 (398015.616, 1251302.340) | 2393.061 (1284.411, 4037.998) | 1838599.827 (990069.769, 3200108.896) | 2499.430 (1345.921, 4350.293) | 4.445 (−4.580, 11.762) | 0.122 (0.086, 0.159) |
| East Asia | 6073716.225 (3033214.164, 11148360.011) | 1319.972 (659.194, 2422.820) | 4105774.425 (1936430.390, 7393810.463) | 1190.247 (561.363, 2143.435) | −9.828 (−16.009, −4.101) | −0.567 (−0.806, −0.327) |
| Eastern Europe | 743548.432 (351658.085, 1353354.075) | 1105.231 (522.714, 2011.663) | 537537.473 (253394.572, 978003.244) | 1164.519 (548.953, 2118.743) | 5.364 (1.958, 10.450) | −0.266 (−0.484, −0.047) |

*(Continued)*

| Location | 1990 | | 2021 | | 1990-2021 | |
|---|---|---|---|---|---|---|
| | Prevalence cases (95%UI) | Prevalence rate(95%UI) | Prevalence cases (95%UI) | Prevalence rate(95%UI) | Rate change (95%UI) | EAPC (95%UI) |
| Eastern Sub-Saharan Africa | 1541400.429 (761821.165, 2757445.126) | 1389.934 (686.960, 2486.483) | 3378554.567 (1668894.239, 6118156.963) | 1484.526 (733.307, 2688.299) | 6.806 (3.606, 10.054) | 0.153 (0.129, 0.176) |
| High-income Asia Pacific | 324809.075 (141833.339, 607342.462) | 645.401 (281.825, 1206.799) | 192950.561 (84191.540, 361602.163) | 626.686 (273.447, 1174.452) | −2.900 (−8.630, 1.997) | −0.063 (−0.111, −0.014) |
| High-income North America | 475256.089 (204295.512, 867721.650) | 581.496 (249.964, 1061.695) | 546047.016 (242337.141, 1020535.850) | 609.713 (270.592, 1139.525) | 4.852 (−0.260, 11.633) | 0.023 (−0.077, 0.123) |
| North Africa and Middle East | 2301191.090 (1151556.313, 4044008.208) | 1301.783 (651.435, 2287.694) | 3151325.985 (1536682.665, 5657800.639) | 1332.532 (649.783, 2392.389) | 2.362 (−6.986, 9.705) | −0.031 (−0.084, 0.023) |
| Oceania | 30667.378 (14712.468, 56234.178) | 910.987 (437.040, 1670.460) | 57721.054 (27179.545, 108183.515) | 903.815 (425.586, 1693.972) | −0.787 (−7.504, 6.424) | −0.015 (−0.047, 0.016) |
| South Asia | 4211731.466 (2007697.673, 7404290.373) | 776.459 (370.131, 1365.027) | 6207827.031 (2931371.542, 10912802.883) | 908.248 (428.880, 1596.619) | 16.973 (11.353, 25.049) | 0.714 (0.627, 0.801) |
| Southeast Asia | 2374700.990 (1155396.030, 4267813.292) | 1079.895 (525.416, 1940.787) | 2546204.922 (1238560.199, 4522132.091) | 1110.563 (540.216, 1972.392) | 2.840 (−0.694, 6.237) | −0.079 (−0.255, 0.098) |
| Southern Latin America | 130978.693 (60348.359, 241619.825) | 675.869 (311.406, 1246.793) | 137709.292 (65041.147, 246876.716) | 705.853 (333.380, 1265.409) | 4.436 (−4.041, 18.007) | 0.102 (0.042, 0.161) |
| Southern Sub-Saharan Africa | 882914.587 (485181.819, 1456201.746) | 3336.705 (1833.596, 5503.268) | 1031703.112 (571705.309, 1685983.456) | 3299.923 (1828.611, 5392.651) | −1.102 (−5.313, 2.399) | −0.442 (−0.551, −0.332) |
| Tropical Latin America | 1088205.393 (533152.287, 1931230.447) | 1571.150 (769.765, 2788.309) | 1055273.681 (523624.038, 1874475.666) | 1584.792 (786.370, 2815.056) | 0.868 (−0.939, 3.510) | −0.080 (−0.189, 0.029) |
| Western Europe | 582689.684 (249804.803, 1054117.468) | 592.492 (254.007, 1071.850) | 536028.780 (229292.253, 974867.436) | 584.477 (250.017, 1062.980) | −1.353 (−5.758, 1.960) | −0.081 (−0.106, −0.057) |
| Western Sub-Saharan Africa | 1558160.555 (790417.343, 2763321.006) | 1449.529 (735.311, 2570.669) | 4075122.408 (2071768.800, 7247141.345) | 1517.306 (771.390, 2698.357) | 4.676 (2.771, 6.626) | 0.129 (0.089, 0.169) |

**Abbreviations:** UI, uncertainty interval. EAPC, estimated annual percentage changes. SDI, Social Development Index.

### Trends in disease burden correlated with SDI

As illustrated in Fig 2, from 1990 to 2021, there was a significant negative correlation between the Socio-Demographic Index (SDI) and the burden of NVL across 21 regions. Specifically, a strong inverse relationship was observed between SDI and the ASPR (Fig 2A, r = −0.6226, P < 0.001), indicating that NVL incidence tends to be higher in regions with lower economic development, weaker healthcare infrastructure, and limited access to preventive measures. Similarly, the ASR of DALYs exhibited a comparable negative correlation with SDI (Fig 2B, r = −0.6199, P < 0.001), reinforcing the notion that socioeconomic disparities contribute significantly to disease burden.

**Table 2. DALYs of near vision loss between 1990 and 2021 at the Global and Regional Level.**

| Location | 1990 DALYs cases (95%UI) | 1990 DALYs rate (95%UI) | 2021 DALYs cases (95%UI) | 2021 DALYs rate (95%UI) | 1990-2021 Rate change (95%UI) | EAPC (95%UI) |
|---|---|---|---|---|---|---|
| Global | 263635.767 (100159.742, 563502.158) | 11.673 (4.435, 24.949) | 330975.887 (125175.589, 706474.220) | 12.557 (4.749, 26.803) | 7.574 (5.335, 10.008) | 0.139 (0.085, 0.194) |
| **SDI** | | | | | | |
| High-middle SDI | 45198.658 (17452.221, 96870.967) | 12.210 (4.715, 26.170) | 34095.740 (12635.189, 73974.498) | 11.239 (4.165, 24.385) | −7.953 (−14.056, −3.208) | −0.617 (−0.812, −0.423) |
| High SDI | 18174.837 (6295.101, 40901.746) | 7.232 (2.505, 16.275) | 16647.428 (5813.198, 37684.402) | 7.153 (2.498, 16.193) | −1.084 (−3.783, 1.528) | −0.179 (−0.261, −0.098) |
| Low-middle SDI | 61702.334 (23268.874, 132133.498) | 10.440 (3.937, 22.357) | 91247.046 (34122.006, 195708.276) | 11.937 (4.464, 25.603) | 14.343 (10.175, 18.456) | 0.513 (0.469, 0.556) |
| Low SDI | 37004.807 (14207.255, 77946.088) | 13.236 (5.082, 27.880) | 86258.113 (33347.180, 181921.721) | 14.765 (5.708, 31.140) | 11.552 (8.698, 14.309) | 0.349 (0.319, 0.379) |
| Middle SDI | 101350.603 (38892.793, 217360.651) | 13.256 (5.087, 28.430) | 102496.605 (39160.872, 221036.042) | 13.681 (5.227, 29.503) | 3.204 (0.420, 6.234) | −0.022 (−0.141, 0.097) |
| **Regions** | | | | | | |
| Andean Latin America | 2622.569 (1010.362, 5679.829) | 13.835 (5.330, 29.963) | 3307.582 (1264.764, 6962.158) | 13.972 (5.343, 29.409) | 0.988 (−7.351, 10.606) | 0.151 (0.110, 0.191) |
| Australasia | 411.814 (141.624, 950.336) | 6.565 (2.258, 15.149) | 478.732 (158.302, 1137.167) | 6.348 (2.099, 15.078) | −3.305 (−13.342, 8.455) | −0.065 (−0.124, −0.006) |
| Caribbean | 2253.763 (884.883, 4813.202) | 14.927 (5.861, 31.878) | 2291.734 (882.482, 4898.523) | 15.015 (5.782, 32.095) | 0.595 (−3.770, 6.562) | 0.047 (−0.020, 0.114) |
| Central Asia | 2925.680 (1052.362, 6409.483) | 9.264 (3.332, 20.296) | 3210.598 (1157.298, 7029.965) | 9.273 (3.342, 20.303) | 0.089 (−4.484, 4.652) | −0.151 (−0.360, 0.058) |
| Central Europe | 2610.466 (934.602, 5932.913) | 6.648 (2.380, 15.109) | 1537.901 (553.412, 3573.771) | 6.528 (2.349, 15.171) | −1.796 (−5.820, 1.500) | −0.260 (−0.363, −0.158) |
| Central Latin America | 12571.685 (4761.439, 26402.443) | 15.214 (5.762, 31.952) | 13606.260 (5204.242, 28864.159) | 15.954 (6.102, 33.844) | 4.862 (−4.803, 10.323) | 0.115 (0.075, 0.154) |
| Central Sub-Saharan Africa | 7677.508 (2968.413, 16470.955) | 24.776 (9.579, 53.152) | 19146.768 (7285.236, 41619.947) | 26.029 (9.904, 56.579) | 5.057 (−4.563, 13.154) | 0.141 (0.103, 0.179) |
| East Asia | 63547.394 (24195.742, 136425.155) | 13.810 (5.258, 29.649) | 43021.164 (16103.282, 94620.843) | 12.472 (4.668, 27.430) | −9.694 (−16.450, −3.601) | −0.562 (−0.801, −0.323) |
| Eastern Europe | 7773.295 (2858.340, 17118.557) | 11.554 (4.249, 25.446) | 5621.248 (2104.088, 12191.038) | 12.178 (4.558, 26.411) | 5.395 (2.089, 10.483) | −0.262 (−0.479, −0.044) |
| Eastern Sub-Saharan Africa | 15986.149 (6070.338, 34174.216) | 14.415 (5.474, 30.816) | 35174.956 (13397.404, 74448.616) | 15.456 (5.887, 32.712) | 7.218 (3.599, 10.610) | 0.169 (0.146, 0.192) |
| High-income Asia Pacific | 3392.146 (1131.336, 7980.835) | 6.740 (2.248, 15.858) | 2014.479 (659.606, 4716.865) | 6.543 (2.142, 15.320) | −2.929 (−8.448, 3.020) | −0.062 (−0.110, −0.013) |
| High-income North America | 4965.097 (1668.802, 11702.612) | 6.075 (2.042, 14.319) | 5684.486 (1950.358, 13204.217) | 6.347 (2.178, 14.744) | 4.482 (−0.748, 12.587) | 0.013 (−0.087, 0.114) |
| North Africa and Middle East | 23993.371 (9203.661, 50211.969) | 13.573 (5.207, 28.405) | 32881.336 (12583.745, 70593.158) | 13.904 (5.321, 29.850) | 2.437 (−7.788, 9.924) | −0.025 (−0.078, 0.028) |
| Oceania | 319.380 (115.933, 703.480) | 9.487 (3.444, 20.897) | 602.109 (218.042, 1317.491) | 9.428 (3.414, 20.630) | −0.625 (−9.495, 9.034) | −0.010 (−0.042, 0.023) |
| South Asia | 43704.001 (15897.982, 97069.765) | 8.057 (2.931, 17.895) | 64649.855 (23473.637, 140727.856) | 9.459 (3.434, 20.589) | 17.396 (11.274, 25.482) | 0.724 (0.637, 0.812) |
| Southeast Asia | 24771.290 (9280.256, 52822.282) | 11.265 (4.220, 24.021) | 26605.151 (9947.971, 57537.830) | 11.604 (4.339, 25.096) | 3.014 (−0.741, 6.647) | −0.071 (−0.247, 0.106) |
| Southern Latin America | 1367.661 (461.056, 3202.023) | 7.057 (2.379, 16.523) | 1437.285 (497.830, 3317.276) | 7.367 (2.552, 17.003) | 4.389 (−6.482, 18.105) | 0.102 (0.042, 0.162) |

*(Continued)*

Table 2. (Continued)

| Location | 1990 | | 2021 | | 1990-2021 | |
|---|---|---|---|---|---|---|
| | DALYs cases (95%UI) | DALYs rate (95%UI) | DALYs cases (95%UI) | DALYs rate (95%UI) | Rate change (95%UI) | EAPC (95%UI) |
| Southern Sub-Saharan Africa | 9194.119 (3743.162, 19971.750) | 34.746 (14.146, 75.477) | 10752.077 (4360.557, 23394.273) | 34.391 (13.947, 74.827) | −1.023 (−5.089, 2.647) | −0.435 (−0.544, −0.326) |
| Tropical Latin America | 11321.301 (4321.672, 23945.369) | 16.346 (6.240, 34.572) | 10981.236 (4214.695, 23416.377) | 16.491 (6.330, 35.166) | 0.892 (−1.624, 3.766) | −0.074 (−0.182, 0.034) |
| Western Europe | 6077.402 (2022.644, 14354.831) | 6.180 (2.057, 14.596) | 5587.421 (1863.731, 13366.770) | 6.092 (2.032, 14.575) | −1.411 (−6.143, 2.804) | −0.081 (−0.105, −0.057) |
| Western Sub-Saharan Africa | 16149.677 (6225.002, 34037.893) | 15.024 (5.791, 31.665) | 42383.508 (16188.391, 89475.072) | 15.781 (6.027, 33.315) | 5.039 (2.877, 7.409) | 0.143 (0.102, 0.183) |

**Abbreviations:** DALYs, Disability-Adjusted Life Years. UI, uncertainty interval. EAPC, estimated annual percentage changes. SDI, Social Development Index.

Further analysis of the SDI distribution among 204 countries in 2021 corroborated these findings. Negative correlations persisted between SDI and NVL-related metrics, with ASPR and ASR of DALYs demonstrating statistically significant inverse relationships (Fig 2C, r = −0.5671, P < 0.001; Fig 2D, r = −0.5654, P < 0.001). These trends underscore the disproportionate impact of NVL in lower-SDI countries, where limited access to early diagnosis, medical interventions, and healthcare resources exacerbates disease prevalence and long-term disability.

The findings emphasize the urgent need for targeted public health strategies, particularly in low- and middle-SDI regions, to mitigate the growing burden of NVL. Strengthening healthcare infrastructure, improving early detection programs, and implementing educational initiatives could play a crucial role in reducing disease incidence and alleviating its impact on affected populations.

## Distribution of disease burden across 204 countries

The geographical distribution of NVL burden, as illustrated in Fig 3A–3D, indicates that low-income countries, particularly in Africa, bear the heaviest disease burden. These regions exhibit notably higher ASPR and ASR of DALYs, reflecting the substantial impact of NVL in resource-limited settings. In contrast, significantly lower ASPR and ASR of DALYs are observed in the Americas, Australia, and Europe (Fig 3A and 3B), where stronger healthcare infrastructure, better access to medical interventions, and preventive measures likely contribute to reduced disease prevalence and burden.

The trends in EAPC for ASPR and ASR of DALYs across the 204 countries are further visualized in Fig 3C and 3D. Notably, while regions with historically lower disease burdens, such as Australia, exhibit a growing trend in NVL prevalence, several developing nations in Africa and Asia are experiencing a sharp and sustained increase in NVL burden. This pronounced upward trajectory underscores the urgent need for targeted public health interventions, including improved access to early diagnosis, enhanced healthcare resources, and strengthened disease prevention programs in these high-risk regions.

## Sex and age dynamics in NVL

Across 21 global regions, the ASPR and ASR of DALYs have consistently been higher among adolescent females than males in both 1990 and 2021, highlighting notable gender disparities in NVL prevalence (Fig 4A–4D). This disparity is particularly pronounced in the Caribbean region, where adolescent females face a disproportionately higher disease burden. The underlying factors contributing to this trend may include biological, behavioral, and healthcare access differences, warranting further investigation into gender-specific risk factors and targeted intervention strategies.

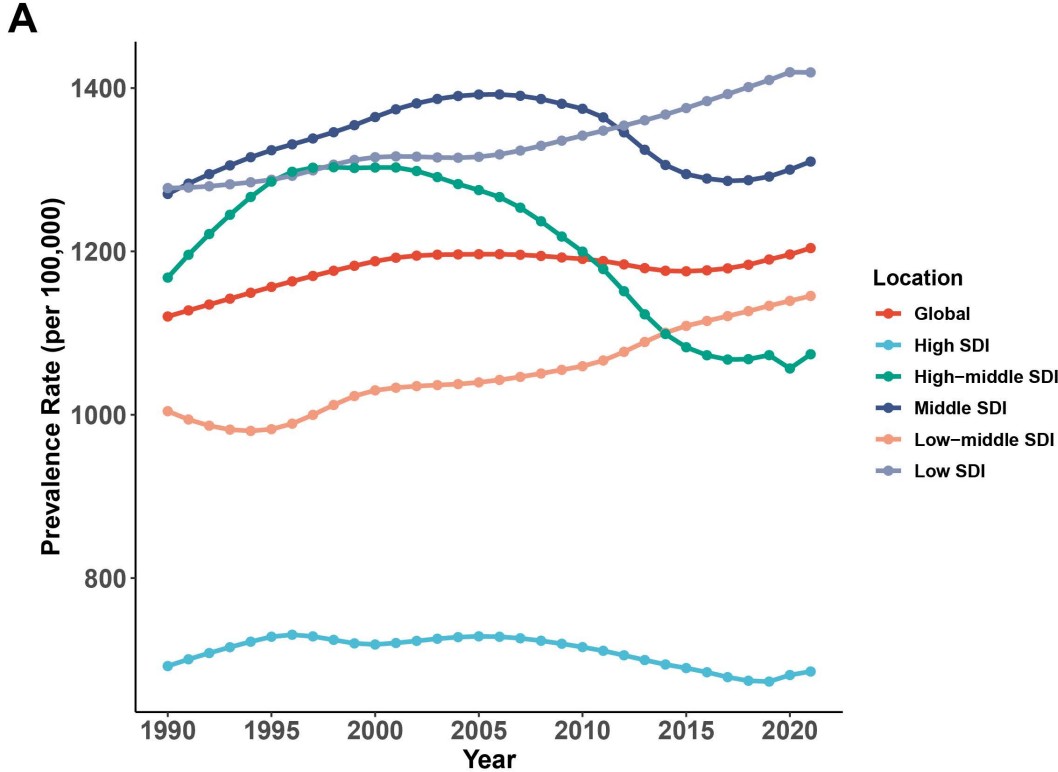

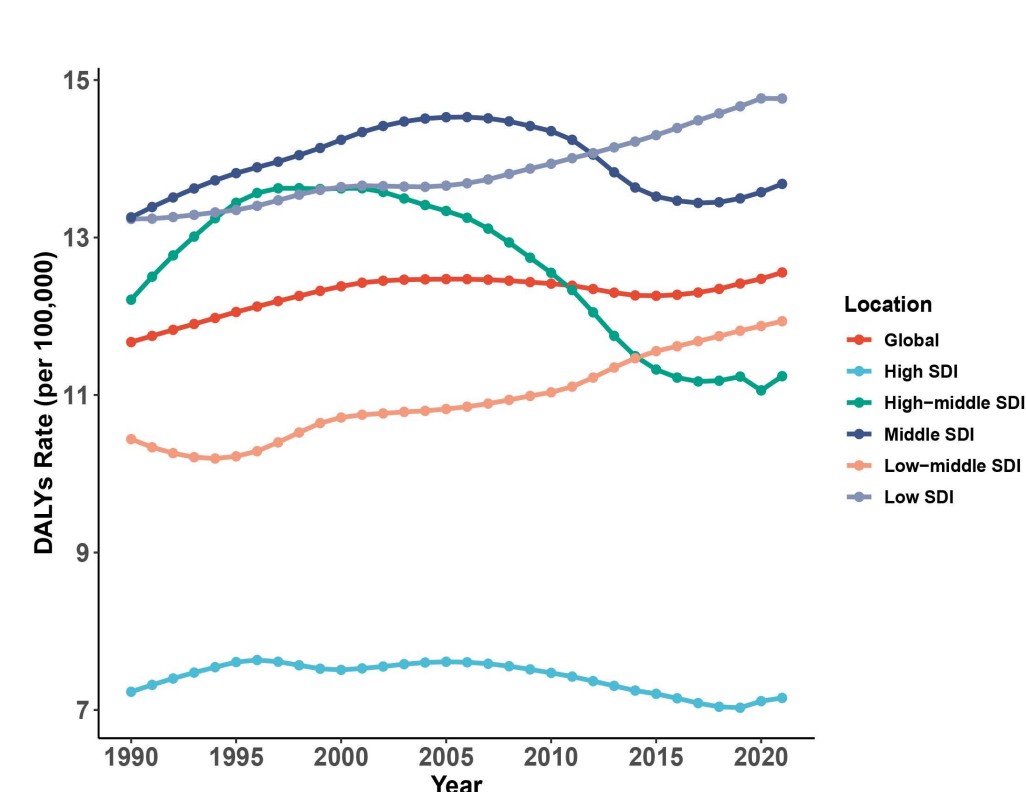

**Fig 1.  The trend of rates including age-standardized prevalence (A) and DALYs (B) of adolescent NVL in global and 5 Socio-Demographic Index (SDI) regions from 1990-2021.**

## Table 3. Glossary of abbreviations.

| Term | Definition |
|------|------------|
| Near vision loss (NVL) | A condition in which the eye has difficulty focusing on near objects, resulting in blurred near vision |
| Global Burden of Disease (GBD) | A research program that quantifies health loss from diseases, injuries, and risk factors to compare and improve health systems globally |
| Disability-Adjusted Life Years (DALYs) | A measure of the burden of diseases, injuries, and premature death on a population |
| Bayesian Age-Period-Cohort (BAPC) | A statistical model used to analyze and predict data by considering the effects of age, period, and cohort |
| Social Development Index (SDI) | A composite indicator of development status strongly correlated with health outcomes |
| Age-standardized rates (ASRs) | A statistical tool used to compare the rates of events between different populations or over time, by adjusting for differences in age structure |
| Age-standardized prevalence rate (ASPR) | A statistical measure used to compare the prevalence of a disease or condition across different populations or over time, by adjusting for differences in age structure |
| Estimated Annual Percentage Change (EAPC) | A statistical measure used to quantify the average annual change in rates, over a specified time period |

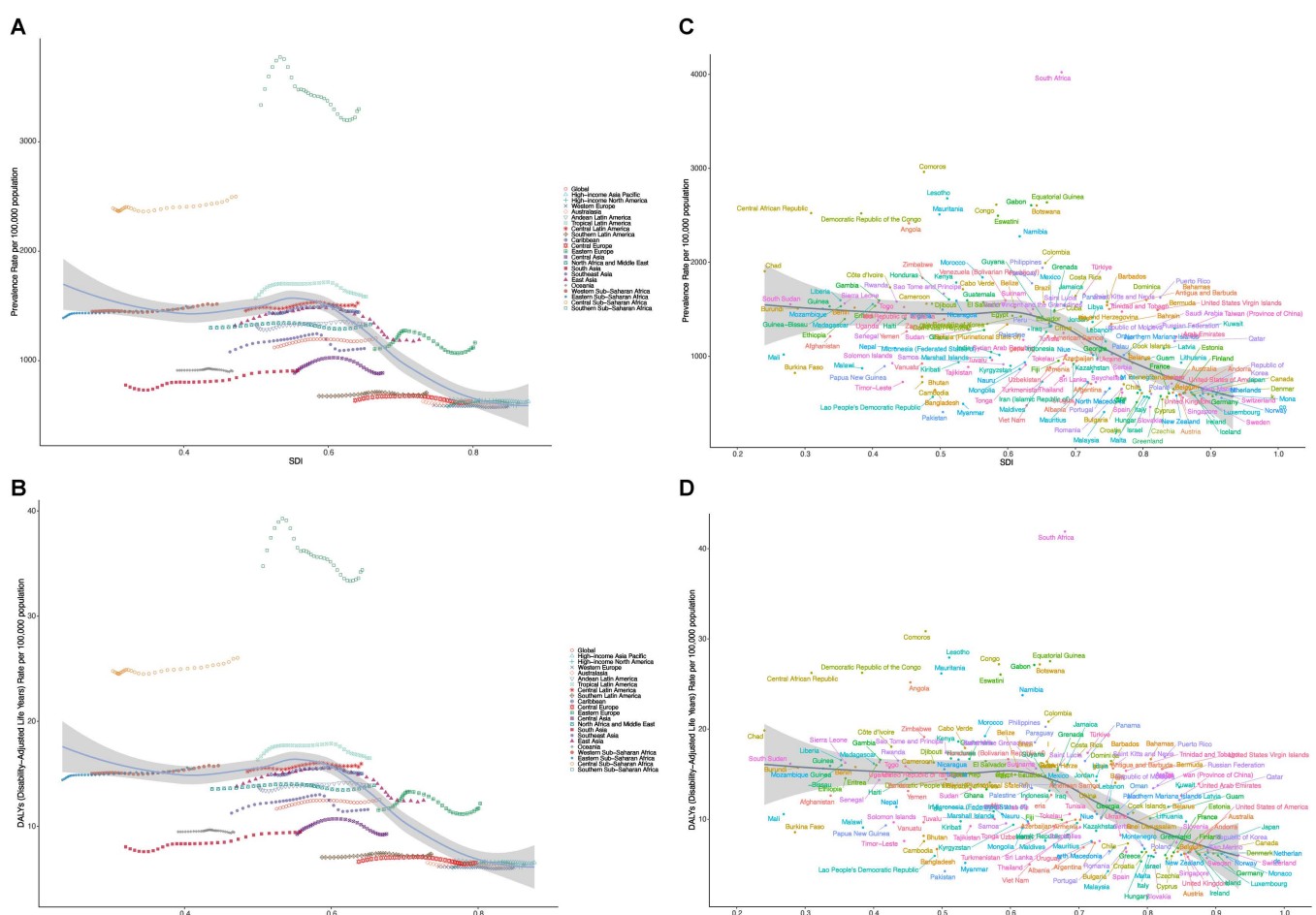

**Fig 2. The trends including age-standardized prevalence (A) and DALYs (B) of adolescent NVL for 21 regions with different Socio-Demographic Index (SDI) values from 1990-2021.** The trends including age-standardized rate of prevalence (C) and DALYs (D) of adolescent NVL for 204 countries with different Socio-Demographic Index (SDI) values in 2021.

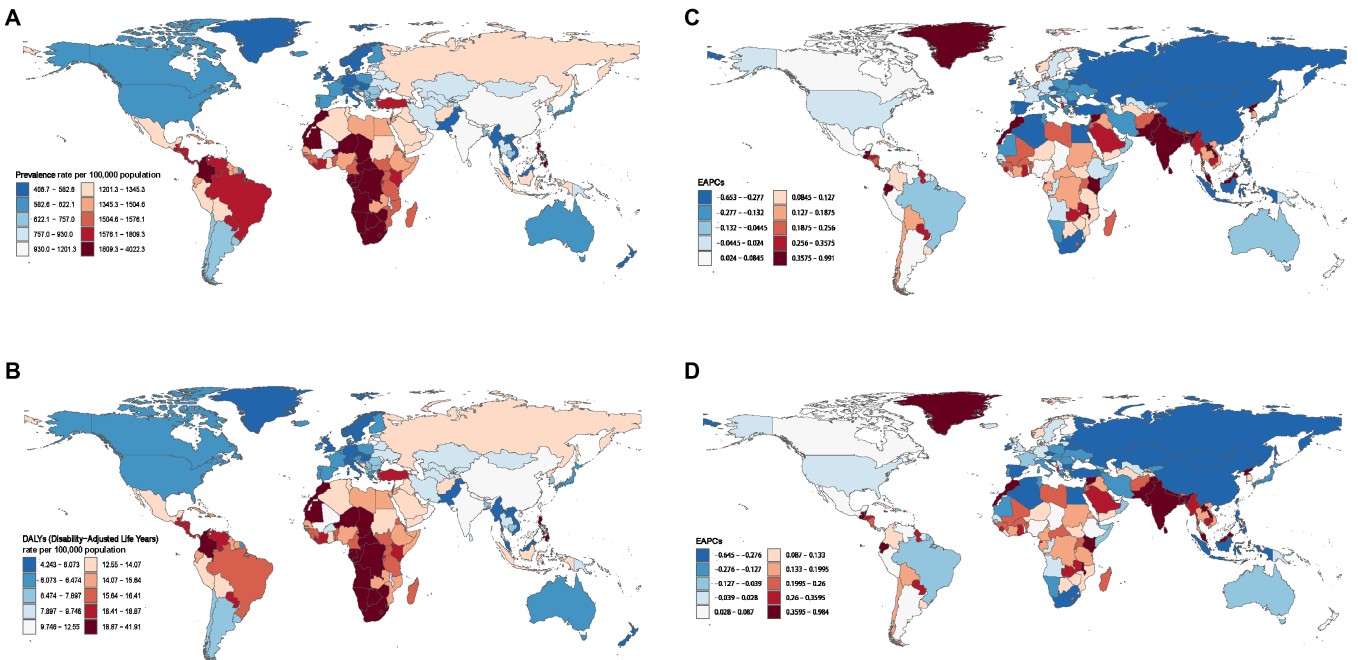

**Fig 3. The age-standardized prevalence (A) and DALYs (B) rate of adolescent NVL in 204 countries in 2021.** The estimated annual percentage changes (EAPC) of age-standardized prevalence (C), and DALYs (D) rate of adolescent NVL in 204 countries from 1990 to 2021.

From 1990 to 2021, the 15–19 years age group had the highest number of NVL cases, making it the most affected demographic. Notably, NVL prevalence has increased across all age groups (Fig 5A–5D), with the most significant rise (7%) occurring in the < 5 years age group (Fig 5A), suggesting a growing concern for early childhood NVL cases. However, in contrast to these trends, the prevalence in the 15–19 years age group remained relatively stable between 1995 and 2015 (Fig 5D), indicating potential age-specific differences in disease progression and risk factors.

These findings emphasize the need for age- and gender-specific approaches to NVL prevention and management, particularly in vulnerable populations such as young children and adolescent females. Targeted public health measures, including early screening, improved healthcare access, and gender-responsive interventions, could help mitigate the rising NVL burden.

## Prediction to 2060

The GBD study provides future projections for NVL prevalence and disease burden through 2060, as illustrated in Fig 6A–6D. Globally, the number of NVL cases is expected to stabilize gradually, with an estimated 33 million children and adolescents (95% UI: 13.45–52.6) affected by NVL by 2060 (Fig 6A).

Despite this stabilization in absolute numbers, the ASPR of NVL is projected to continue increasing (Fig 6B). By 2060, the ASPR is expected to reach 1,433 cases per 100,000 population (95% UI: 583.75–2,282.35), indicating that the proportion of affected individuals may steadily rise over time.

Similarly, the ASR of DALYs for NVL is also projected to continue its upward trend. By 2060, the ASR is estimated to reach 14.98 per 100,000 population (95% UI: 5.73–24.23, Fig 6D), reflecting a sustained disease burden despite potential advances in healthcare interventions.

These projections underscore the need for sustained global efforts to mitigate the rising NVL burden, particularly through early intervention strategies, improved healthcare access, and targeted public health policies aimed at reducing NVL prevalence and its long-term impact.

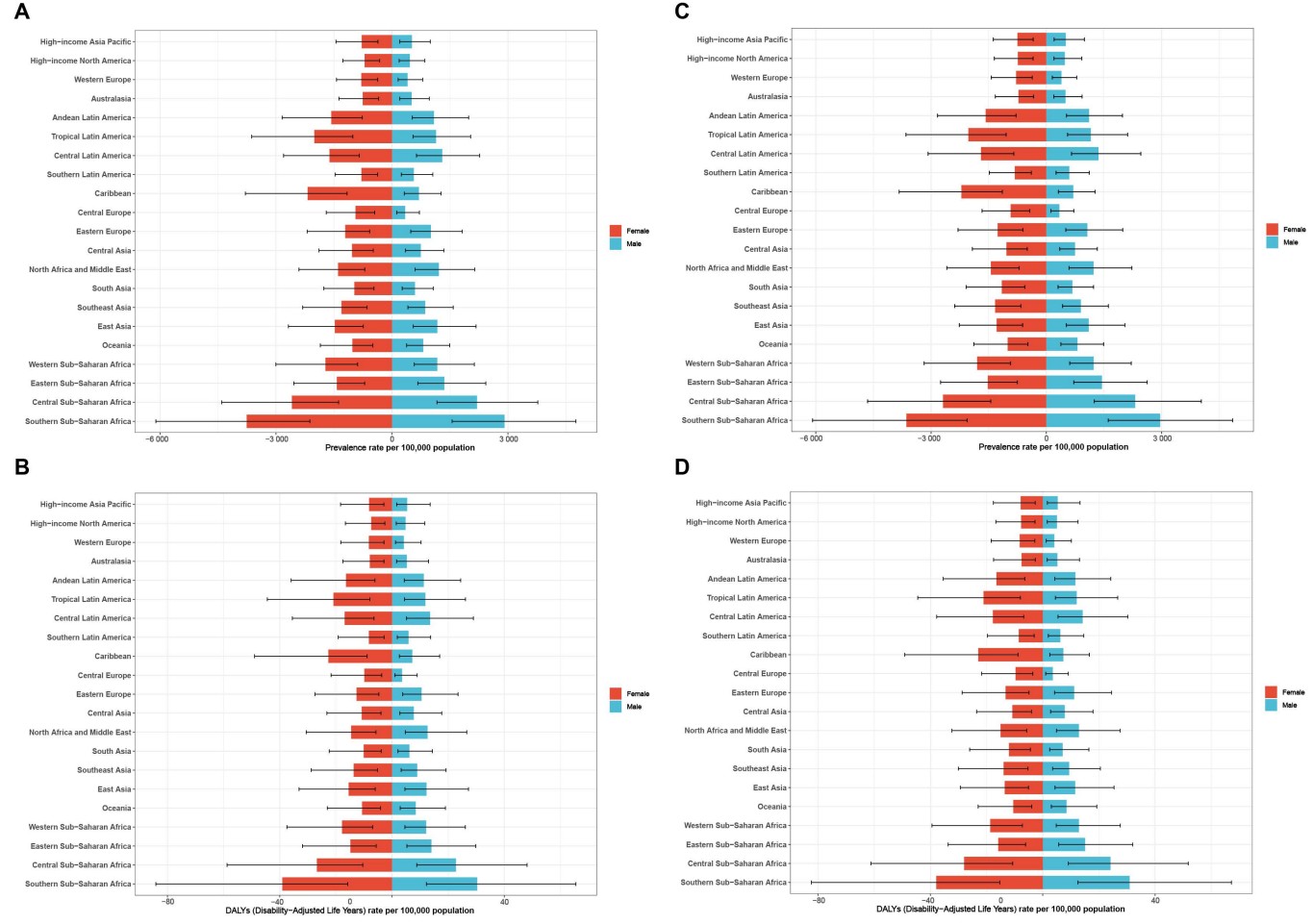

**Fig 4. The difference about age-standardized prevalence (A) and DALYs (B) rate of adolescent NVL between females and males across 21 regions in 1990.** The difference about age-standardized prevalence (C) and DALYs (D) rate of adolescent NVL between females and males across 21 regions in 2021.

## Discussion

This study examines global trends in the prevalence and disease burden of NVL among children and adolescents under 20 years of age. From 1990 to 2021, the global prevalence of NVL in this demographic increased by 25.3%, indicating a gradual upward trend. After adjustment, the ASPR and ASR of DALYs exhibited a declining trend between 2000 and 2015, with higher SDI regions contributing most significantly to this reduction. The disease burden of NVL varied across SDI regions, with low-SDI regions, particularly in Africa, experiencing the highest burden. Gender-specific analysis revealed that age-specific prevalence and DALY rates were generally higher in females than in males. By age group, individuals aged 15–19 years consistently represented the largest proportion of cases; however, the fastest growth in ASPR was observed among children under 5 years of age, indicating an increasing impact of NVL in younger populations. Based on our prediction model, the global ASPR for NVL is expected to increase slowly in the future, while overall prevalence is projected to remain relatively stable. Nevertheless, the global burden of NVL among children and adolescents under 20 years of age is anticipated to persist at high levels.

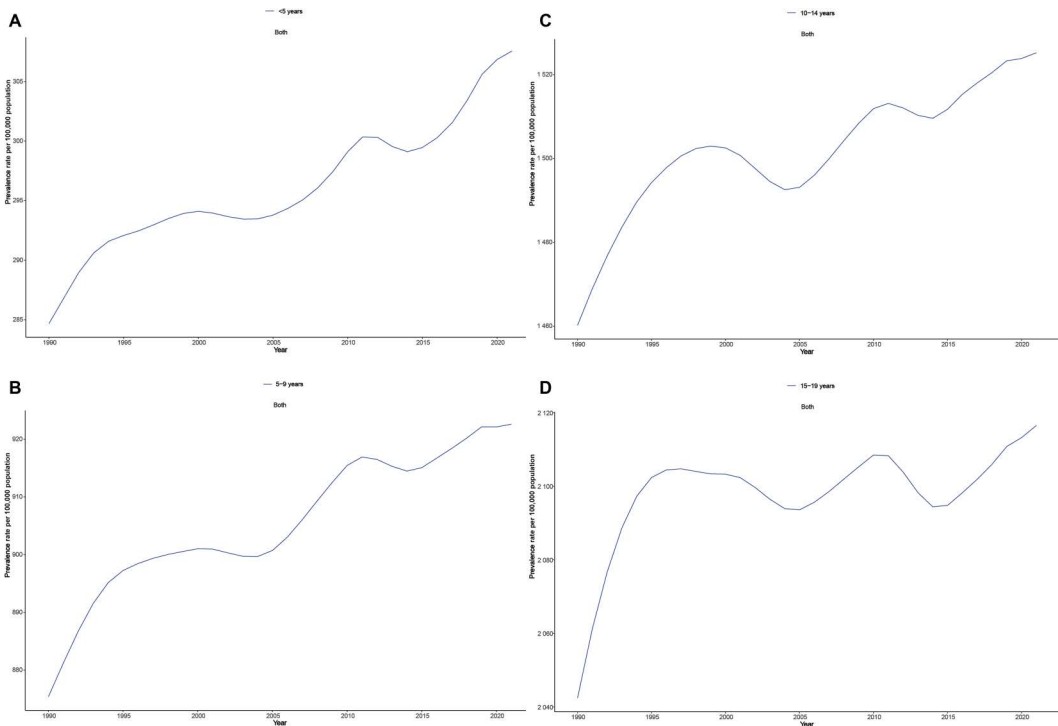

**Fig 5. The rates including age-standardized prevalence of adolescent NVL for different age groups globally in 2021.**

The higher rate of increase in the global burden of NVL compared to previous studies that did not restrict age is understandable. First, past studies have shown that the burden of NVL primarily affects individuals aged 45–80 [22,23]. Second, increasing global aging has led to a larger base and higher proportion of older people [24]. Although NVL is currently closely associated with aging, it is also increasingly affecting younger populations [25]. Myopia, one of the most common and fastest-growing diseases globally, is driven by factors such as reduced outdoor time, increased near-work activities, and rising urbanization in developing countries [26,27]. Uncorrected or high levels of myopia can lead to NVL through various pathways, including myopic macular degeneration or its complications, cataracts, retinal detachment, and glaucoma [28]. There is currently a large global base of myopia patients, with adolescents comprising a significant proportion. This trend is particularly concerning. For instance, in East and Southeast Asia, the prevalence of myopia among 17- to 18-year-olds is 80–90% [29]. Data from the current GBD study reveal that these regions have high EAPC values for ASPR and ASDR in 2021, further highlighting the need to address NVL in children and adolescents. In our study, children and adolescents under the age of 20 years exhibited distinct trends in disease burden across various SDI regions and ages. After adjusting for age and SDI separately, we found that during the period from 2000 to 2015, children and adolescents in higher SDI regions experienced a yearly decline in ASPR. This decline may be related to the 1999 WHO release of VISION 2020, which set the goal of eliminating avoidable blindness by 2020 [30].

Concurrently, the global ASPR for children and adolescents aged 15–19 years remained stable during this period. This stability may be attributed to the year-on-year increases in the ASPR among children and adolescents in lower SDI regions. These increases are likely primarily driven by disparities in economic levels. For example, the burden of disease among adolescents under 20 years of age in the South African region is particularly striking in our findings. Nearly 30% of the world's Multidimensional Poverty Index (MPI) population lives in South Africa, and the South African region has some of the lowest levels of infrastructure investment in the world [31]. In economically underdeveloped areas, the burden of

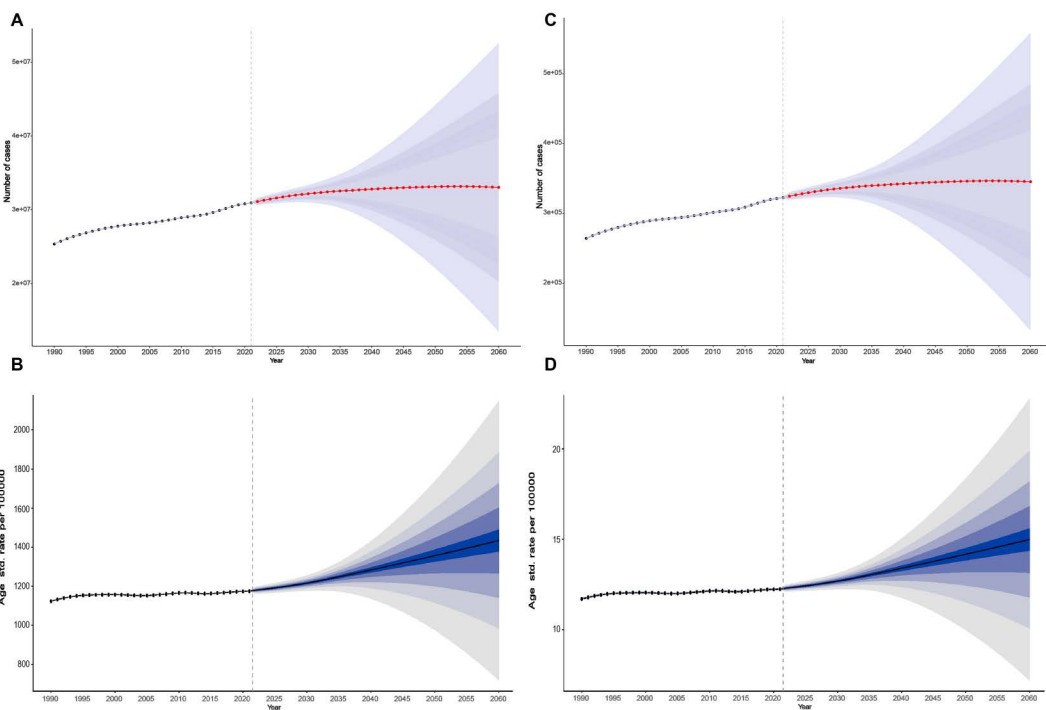

**Fig 6. The trend of cases (A) and age-standardized rate (B) of prevalence in adolescent NVL from 1990-2021 and prediction to 2060 in global.**
The trend of cases (C) and age-standardized rate (D) of DALYs in adolescent NVL from 1990-2021 and prediction to 2060 in global.

NVL is particularly pronounced due to limited awareness and access to eye care services, insufficient numbers of health-care facilities and professionals, and a lack of available and affordable eyewear [32,33]. Given the persistence of socio-economic inequalities, future research should prioritize the development and implementation of more effective strategies, the establishment of more equitable and universal goals and targets, and a focused examination of the disease burden in economically underdeveloped regions.

Compared to previous studies, our analysis revealed a reversal of the previously observed downward trend in adolescent ASPR in higher SDI regions after 2020. This reversal may be attributed to the global outbreak of COVID-19, which began in 2020. The impact of COVID-19 on children and adolescents under the age of 20 years with NVL is multifaceted. First, the pandemic has exacerbated the shortage of resources for ophthalmic care. During India's nationwide lockdown, 72.5% of ophthalmic institutions completely halted all clinical work, and elective eye surgeries were canceled as hospitals provided only emergency and critical care services [34,35]. The COVID-19 pandemic also disrupted myopia management strategies due to a significant shift in the schooling and learning patterns of children and adolescents. This shift led to a rapid increase in the number of myopes, driven by excessive use of digital devices and reduced time spent outdoors [36,37]. Additionally, pediatric ophthalmologists faced significant economic challenges during the pandemic. A survey of 243 pediatric ophthalmologists found that 20.9% planned for early retirement due to COVID-19, and 4.7% sold their practices [38]. These findings suggest that the COVID-19 outbreak has led to a marked increase in the incidence and progression of NVL among children and adolescents under the age of 20 years, a trend not captured in previous studies.

This study acknowledges several limitations. Firstly, the database does not provide comprehensive data on the temporal changes in the burden of NVL among children and adolescents under the age of 20 years. The absence of detailed information on NVL risk factors further precluded a thorough risk factor analysis. Secondly, our projections, derived from NVL estimates spanning 1990–2021, did not account for potential NVL control interventions that may be implemented

between now and 2060. This omission may introduce bias and error into our projections. Importantly, the GBD 2021 dataset does not stratify NVL by etiology (e.g., uncorrected refractive error vs. amblyopia) or distance vision status. While our analysis adheres to the GBD's standardized definition of NVL, future studies would benefit from granular data to disentangle the contributions of distinct ocular pathologies. What's more, it is also possible that the increased burden of NVL may stem from population growth. Even if the prevalence of NVL remains constant, an expanding population base may result in an upward trend in the number of people with NVL. In addition, increased screening and surveillance efforts, as well as upgraded diagnostic techniques, have increased the detection rate of NVL, which may be contributing to the rising burden of NVL. Therefore, our findings should be interpreted with caution. Future research endeavors should prioritize the development of more accurate and inclusive projection models. Acknowledging these limitations is essential for the accurate interpretation of GBD research results and the formulation of effective public health policies. Consequently, researchers and policymakers must consider these factors to accurately assess and address the global challenges associated with NVL in children and adolescents.

## Conclusions

In summary, NVL is still a growing public health issue among children and adolescents, projected to affect 33 million individuals by 2060. The COVID-19 pandemic has exacerbated this trend, especially in higher SDI regions. The highest burden is seen in low-SDI regions, particularly among children under 5 years, while females consistently exhibit higher NVL rates than males. Urgent and targeted public health strategies are needed to address these disparities and mitigate the impact of NVL.

## Supporting information

**S1 Fig. The rates including age-standardized prevalence of adolescent NVL for different age groups globally in 2021.**
(PDF)

## Author contributions

**Conceptualization:** Jing Peng, Xi-Yuan Zhou.

**Data curation:** Jing Peng, Cong Zhang.

**Formal analysis:** Jing Peng, Cong Zhang.

**Investigation:** Jing Peng, Xingying Li.

**Methodology:** Jing Peng, Xingying Li.

**Supervision:** Xi-Yuan Zhou.

**Validation:** Jin Tang.

**Writing – original draft:** Jing Peng, Xi-Yuan Zhou.

**Writing – review & editing:** Jing Peng, Xi-Yuan Zhou.

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
