## [Decision Letter · Decision Letter 0]

Dear Dr. Zhou,

We look forward to receiving your revised manuscript.

Kind regards,

Ali Faramarzi, MD, MPH

Academic Editor

PLOS ONE

Journal Requirements:

Reviewers' comments:

Reviewer's Responses to Questions

**Comments to the Author**

1. Is the manuscript technically sound, and do the data support the conclusions?

Reviewer #1: Yes

Reviewer #2: Partly

Reviewer #3: Yes

Reviewer #4: Yes

2. Has the statistical analysis been performed appropriately and rigorously?

Reviewer #1: Yes

Reviewer #2: I Don't Know

Reviewer #3: Yes

Reviewer #4: Yes

3. Have the authors made all data underlying the findings in their manuscript fully available?

Reviewer #1: Yes

Reviewer #2: Yes

Reviewer #3: Yes

Reviewer #4: Yes

4. Is the manuscript presented in an intelligible fashion and written in standard English?

Reviewer #1: Yes

Reviewer #2: Yes

Reviewer #3: Yes

Reviewer #4: Yes

Reviewer #1: Thank you for submitting this paper for review, my comments are below. I have some comments relate dot content and a few more related to plot presentation.

I look forward to seeing your updated manuscript.

Introduction

“The COVID-19 pandemic has further exacerbated the situation. Lockdowns and social

distancing measures led to a dramatic increase in screen time and a reduction in outdoor activities,”

Please provide a reference here

The introduction section needs to include what the disease burden actually is, and why we should do something about it. Your paper rightly indicates that NVL is a problem and that something should be done bc its getting worse. But what does it actually mean for people, what is the burden.

You should also include a little more on how NVL comes about what what drives it. You discuss this in a very broad way, but this needs clarity.

Statistical Analysis

“To ensure accurate analysis, the “broom” and “dplyr” packages were loaded. These tools were crucial for efficiently extracting and organizing regression results,”

Consider instead “The “broom” and “dplyr” packages were used, as they were crucial for accurate analysis, efficiently extracting and organizing regression results….”

Tables – consider spaces after commas and parentheses for readability. i.e., 1739897.006 (805939.806, 3085387.571), not 1739897.006(805939.806,3085387.571)

Figure 1 – Consider standardising Y-axis on all plots. Currently, the differing Y-Max makes the plots less instructive than they could be

Global trends and variations across SDI regions

“In high, high-middle, and middle SDI regions, the age-standardized prevalence rate (ASPR) and age-standardized DALY rate (ASR of DALYs) initially peaked before experiencing a decline.”

Consider adding the years to this sentence to better orient the reader to peaks/declines.

“Of particular concern, low-middle SDI regions demonstrated the most rapid increase in disease burden. Between 1990 and 2021, prevalence and DALYs in these regions rose by 14.045% and 14.343%, respectively (Tables 1 & 2).”

Consider “Of particular concern, low-middle SDI regions demonstrated the most rapid increase in disease burden. Between 1990 and 2021, [the change in] prevalence and DALYs in these regions rose by 14.045% and 14.343%, respectively (Tables 1 & 2).”

Trends in Disease Burden Correlated with SDI

Your comment on the outlier of South Africa?

Figures 3C & D – the pink is hard to distinguish, consider a different colour choice. Figs 3 A&B are much easier to understand.

Figure 4 – Great plots, but for me to compare and contrast 1990 to 2021/see change over time, I find that quite hard. Not only is it difficult to estimate difference over time, but mentally comparing one region to another, the regions are in a different order on the Y-Axis (nice descending order, I get it). But as it stands, I think you can improve this plot to deepen reader understanding. Consider standardising the order of the Y-axis and potentially stacking the regions timewise to make it easier to compare the plots?

Figure 5 – standardise Y-Axis or overall all ages/lines on one plot.

Figure 6 - Standardise text size and correct aspect ratio of plots B & D. Also, Agestd. should probably read Age std.

Discussion

“Although NVL is currently closely associated with aging, it is also increasingly affecting younger populations22. Myopia, one of the most common and fastest-growing diseases globally, is driven by factors such as reduced outdoor time, increased near-work activities, and rising urbanization in developing countries23,24. Uncorrected or high levels of myopia can lead to NVL through various pathways, including myopic macular degeneration or its complications, cataracts, retinal detachment, and glaucoma25.

This information should be included in the introduction section.

Reviewer #2: Thank you for the opportunity to review this manuscript.

Strengths: This manuscript offers a fresh perspective on visual impairment by focusing on near vision loss, a topic that receives less attention compared to distance vision loss, particularly in the pediatric population. This unique angle adds value and interest to readers.

The study provides detailed insights into near vision loss across various demographic regions and includes projections up to 2060, which are particularly valuable.

Main Considerations:

1. The manuscript does not define what "near vision loss" specifically means when extracting results from the Global Burden of Disease (GBD) Study. The reviewer found this definition: “presenting near vision worse than N6 or N8 at 40 cm, while maintaining best-corrected distance visual acuity of 6/12 or better.” If this is the definition used, near vision loss refers strictly to uncorrected presbyopia. However, if the definition includes only near vision worse than N6 or N8 at 40 cm, it may encompass individuals with distance vision loss as well. In pediatric populations, significant refractive errors (hyperopia, myopia, astigmatism), amblyopia, and other ocular conditions are the primary causes of near vision loss. To enhance the manuscript’s relevance, it is important to stratify or account for distance vision loss. For instance:

o Children with amblyopia may experience both distance and near vision loss.

o Children with mild to moderate myopia might face distance vision loss but not near vision loss.

o Children with moderate to high hyperopia may not have distance vision loss but could experience near vision loss.

2. Clarification is needed regarding the choice of age 20 as the upper limit for the study, especially considering the focus on early intervention. The reviewer noted that the GBD results website offers different age ranges and would like to see the rationale for selecting this specific range.

3. The manuscript appears to center its discussion of near vision loss on the increase in myopia and its environmental risk factors. While this is a valid contributing factor, it may not represent the entire picture. The authors should address other potential reasons for the observed increase in near vision loss, such as population growth or improved reporting of cases.

4. Reporting prevalence in terms of case numbers rather than as a rate (e.g., %) is misleading. Throughout the manuscript, prevalence cases are discussed in absolute numbers. Prevalence is typically presented as a rate, and revising this would improve clarity and accuracy.

5. The manuscript uses many abbreviations, which can hinder readability. It is recommended to limit abbreviations to widely recognized terms or include a table listing all abbreviations and their meanings to help readers follow the content more easily.

6. Some figures lack units on the axes. For example, graphs reporting prevalence cases do not specify units, which makes the data presentation unclear. Additionally, presenting prevalence over time with only case numbers can be misleading and should be corrected.

Reviewer #3: In this manuscript the authors have presented a clear picture of the global burden of near vision loss and its future projections in the age group of under 20 years.

The authors use “australasia” in the abstract and tables. I am unaware of such a regional term. Please define the term or edit it to a more familiar term.

In the abstract “Notably, NVL burden in higher SDI regions rebounded post-COVID-19, reversing previous declining trends” seems to be at odds with the rest of the manuscript and the fact that known risk factors such as indoor time increased during the COVID-19 pandemic.

Reviewer #4: This study analyzes the global, regional, and national burden of NVL from 1990 to 2021 and projects future trends up to 2060 using data from the Global Burden of Disease (GBD) Study 2021. The results indicate significant increase in NVL cases, rising to 31.7 million in 2021, with projections reaching 33 million by 2060. A strong negative correlation was observed between the Social Development Index (SDI) and NVL burden. NVL burden in higher SDI regions rebounded post-COVID-19, reversing previous declining trends. The results also indicate the COVID-19 outbreak has led to a marked increase in the incidence and progression of NVL among children and adolescent under the age of 20 years, a trend not captured in previous studies. This study reveals the characteristics and severity of NVL in the post-COVID-2019 era, providing critical data-driven insights for global health strategies. It highlights how the COVID-19 pandemic exacerbated NVL trends through increased screen exposure and reduced outdoor activities, while emphasizing the urgent need for equitable interventions to address disparities in disease burden across socioeconomic contexts. The manuscript should be polished by a native speaker to make it more fluent and sound.

**Do you want your identity to be public for this peer review?** For information about this choice, including consent withdrawal, please see our Privacy Policy

Reviewer #1: No

Reviewer #2: **Yes: ** Anh V Bui

Reviewer #3: No

Reviewer #4: No

---

## [Author Response · Author response to Decision Letter 1]

9 May 2025

Point-by-point response to reviewers

Dear Editor and Reviewers,

Thank you very much for your valuable comments and suggestions, which have greatly helped us to improve the quality of our manuscript. We have carefully considered each of the points raised and made the necessary revisions to address them.

Reviewer #1: Thank you for submitting this paper for review, my comments are below. I have some comments relate dot content and a few more related to plot presentation.I look forward to seeing your updated manuscript.

Comment 1:“The COVID-19 pandemic has further exacerbated the situation. Lockdowns and social distancing measures led to a dramatic increase in screen time and a reduction in outdoor activities,” Please provide a reference here.

Response: Thank you very much for your comments. We have added the relevant references here and separated this sentence from the rest of the text (Line56-58�Reference 9-10).

Comment 2: The introduction section needs to include what the disease burden actually is, and why we should do something about it. Your paper rightly indicates that NVL is a problem and that something should be done bc its getting worse. But what does it actually mean for people, what is the burden. You should also include a little more on how NVL comes about what what drives it. You discuss this in a very broad way, but this needs clarity.

Response: Thank you very much for your suggestions. As you requested, we have provided a more specific description of the economic burden imposed by NVL, as well as the causes and enablers of NVL (Line 41-50).

Comment 3:“To ensure accurate analysis, the “broom” and “dplyr” packages were loaded. These tools were crucial for efficiently extracting and organizing regression results,”

Consider instead “The “broom” and “dplyr” packages were used, as they were crucial for accurate analysis, efficiently extracting and organizing regression results….”.

Response: Thank you for your constructive suggestion. As requested, we have revised the sentence accordingly (Lines 143-144).

Comment 4: Tables – consider spaces after commas and parentheses for readability. i.e., 1739897.006 (805939.806, 3085387.571), not 1739897.006(805939.806,3085387.571)

Response: Thank you for your constructive suggestion. For the numerical parts of the Tables 1&2, we leave spaces after commas and brackets, as you requested.

Comment 5: Figure 1 – Consider standardising Y-axis on all plots. Currently, the differing Y-Max makes the plots less instructive than they could be.

Response: Thank you very much for your helpful suggestion. As you pointed out, using a standardized Y-axis makes the data presentation more convincing. Accordingly, we have redrawn and adjusted the figures based on your advice. Presenting the results of all groups together not only facilitates direct comparison but also enhances the clarity, consistency, and overall visual quality of the figures.

Comment 6: “In high, high-middle, and middle SDI regions, the age-standardized prevalence rate (ASPR) and age-standardized DALY rate (ASR of DALYs) initially peaked before experiencing a decline.”

Consider adding the years to this sentence to better orient the reader to peaks/declines.

Response: Thank you very much for your suggestions. We have added a time note to this sentence (Line 157-160).

Comment 7: “Of particular concern, low-middle SDI regions demonstrated the most rapid increase in disease burden. Between 1990 and 2021, prevalence and DALYs in these regions rose by 14.045% and 14.343%, respectively (Tables 1 & 2).”

Consider “Of particular concern, low-middle SDI regions demonstrated the most rapid increase in disease burden. Between 1990 and 2021, [the change in] prevalence and DALYs in these regions rose by 14.045% and 14.343%, respectively (Tables 1 & 2).”

Response: Thank you very much for your suggestions. We have changed the sentence to the one you suggested (Line 169-171).

Comment 8: Trends in Disease Burden Correlated with SDI.

Your comment on the outlier of South Africa?

Response: Your suggestions are highly constructive. Our comments on the South African region have been added to the Discussion section �Line284-288�. Nearly 30% of those in the World’s Multidimensional Poverty Index (MPI) live in South Africa(Reference 31). South Africa has one of the lowest levels of infrastructure investment in the world, and these deficiencies are particularly evident in health and eye health.

Comment 9: Figures 3C & D – the pink is hard to distinguish, consider a different colour choice. Figs 3 A&B are much easier to understand.

Response: Thank you very much for your valuable suggestion. As you rightly pointed out, Figures 3A and 3B are much easier to understand. Therefore, we have revised the color scheme of Figures 3C and 3D to match the same format, making them easier to read and interpret.

Comment 10: Figure 4 – Great plots, but for me to compare and contrast 1990 to 2021/see change over time, I find that quite hard. Not only is it difficult to estimate difference over time, but mentally comparing one region to another, the regions are in a different order on the Y-Axis (nice descending order, I get it). But as it stands, I think you can improve this plot to deepen reader understanding. Consider standardising the order of the Y-axis and potentially stacking the regions timewise to make it easier to compare the plots?

Response: Thank you very much for your valuable suggestions. As you pointed out, in Fig 4, we prioritized a nice descending order over the standardization of the Y-axis, which might have added some difficulty for readers. Therefore, we have updated the standardized Y-axis order of the figures according to your advice. Additionally, the suggestion you made about stacking the regions timewise is indeed excellent. However, in this particular section, the gender differences between males and females are more pronounced than the differences between 1990 and 2021. Moreover, there are still very few studies in the existing literature reporting epidemiological differences in NVL caused by gender. Our study, therefore, holds greater significance and value. For these reasons, we would like to request to retain the comparison between males and females. You can see the revised figure in Fig 4.

Comment 11: Figure 5 – standardise Y-Axis or overall all ages/lines on one plot

Response: Thank you very much for your high-quality suggestions. Your advice to standardize the Y-axis does indeed make the changes across different ages more evident. However, after careful analysis, we found that the changes across different ages from 1990 to 2021 are relatively minor. Standardizing the Y-axis for all age groups would mask these subtle differences. We have successfully drawn the figures with a unified standardized Y-axis and placed them in the supplementary materials (Fig S1). Given that the changes in Fig S1 are not prominent, we would like to request retaining Fig 5.

Comment 12: Figure 6 - Standardise text size and correct aspect ratio of plots B & D. Also, Agestd. should probably read Age std.

Response: Thank you very much for your valuable and detailed suggestions. We have now readjusted the text to the standard size and corrected the aspect ratio of plots B & D as you requested. In addition, we have also revised the content of the text, changing “Agestd” to “Age std”. You can see the revised figure in Fig 6.

Comment 13: “Although NVL is currently closely associated with aging, it is also increasingly affecting younger populations. Myopia, one of the most common and fastest-growing diseases globally, is driven by factors such as reduced outdoor time, increased near-work activities, and rising urbanization in developing countries. Uncorrected or high levels of myopia can lead to NVL through various pathways, including myopic macular degeneration or its complications, cataracts, retinal detachment, and glaucoma.

This information should be included in the introduction section.

Response: Thank you very much for your suggestions. We have added the above you requested in the Introduction section (Line 41-50).

Reviewer #2: Thank you for the opportunity to review this manuscript.

Strengths: This manuscript offers a fresh perspective on visual impairment by focusing on near vision loss, a topic that receives less attention compared to distance vision loss, particularly in the pediatric population. This unique angle adds value and interest to readers.The study provides detailed insights into near vision loss across various demographic regions and includes projections up to 2060, which are particularly valuable.

Comment 1: The manuscript does not define what "near vision loss" specifically means when extracting results from the Global Burden of Disease (GBD) Study. The reviewer found this definition: “presenting near vision worse than N6 or N8 at 40 cm, while maintaining best-corrected distance visual acuity of 6/12 or better.” If this is the definition used, near vision loss refers strictly to uncorrected presbyopia. However, if the definition includes only near vision worse than N6 or N8 at 40 cm, it may encompass individuals with distance vision loss as well. In pediatric populations, significant refractive errors (hyperopia, myopia, astigmatism), amblyopia, and other ocular conditions are the primary causes of near vision loss. To enhance the manuscript’s relevance, it is important to stratify or account for distance vision loss. For instance:

o Children with amblyopia may experience both distance and near vision loss.

o Children with mild to moderate myopia might face distance vision loss but not near vision loss.

o Children with moderate to high hyperopia may not have distance vision loss but could experience near vision loss.

Response: We sincerely thank you for raising this critical point and for suggesting stratified analyses to improve the clinical relevance of our study. The GBD 2021 Study defines near vision loss (NVL) as *presenting near vision worse than N6 or N8 at 40 cm, with best-corrected distance visual acuity preserved at 6/12 or better*. This definition intentionally excludes individuals with concurrent distance vision loss (<6/12) to isolate near-vision-specific impairment. We have incorporated the definition of this disease into the Methods section (Line 84-89) as per your suggestion.

However, as you astutely notes, pediatric NVL may arise from heterogeneous etiologies (e.g., hyperopia, astigmatism, amblyopia), and stratifying cases by cause or distance vision status would indeed enhance clinical interpretability. Unfortunately, the GBD 2021 database does not provide subgroup data for NVL (e.g., etiology-specific prevalence, concurrent distance vision loss status). This limitation precludes our ability to perform the suggested stratification.

To address this gap, we have formally contacted the GBD collaborative team via email to advocate for future iterations of the GBD database to include etiological or phenotypic stratification of NVL. Such improvements would greatly empower researchers to design targeted interventions, particularly for pediatric populations. We have added the following clarification to the Discussion section (Line317-320).

Comment 2: Clarification is needed regarding the choice of age 20 as the upper limit for the study, especially considering the focus on early intervention. The reviewer noted that the GBD results website offers different age ranges and would like to see the rationale for selecting this specific range.

Response:We thank you for highlighting this important point. The age limit of 20 years was selected based on two key considerations:

Developmental and Behavioral Factors: Vision development stabilizes by late adolescence, and early intervention (e.g., refractive correction, amblyopia therapy) is most effective before age 20. Prolonged screen use and academic demands during school years (ages 5–20) disproportionately increase near-work strain in this group.

GBD Age Stratification: The GBD Study categorizes pediatric data into standardized age groups (e.g., <5, 5-9, 10–14, 15–19 years), with 20 years marking the transition to adult health metrics. Aligning with this structure ensures comparability across GBD-based studies.Now, We have added the following rationale to the Methods section (Line 90-95).

Comment 3: The manuscript appears to center its discussion of near vision loss on the increase in myopia and its environmental risk factors. While this is a valid contributing factor, it may not represent the entire picture. The authors should address other potential reasons for the observed increase in near vision loss, such as population growth or improved reporting of cases.

Response: Thank you very much for your meaningful suggestions. We explore these reasons in the Discussion section (Line 321-325) and describe the reasons why they may contribute to the NVL burden.

Comment 4: Reporting prevalence in terms of case numbers rather than as a rate (e.g., %) is misleading. Throughout the manuscript, prevalence cases are discussed in absolute numbers. Prevalence is typically presented as a rate, and revising this would improve clarity and accuracy.

Response: We sincerely appreciate the your attention to methodological rigor. In alignment with the GBD Study’s standardized reporting framework, age-standardized prevalence rates (ASPR) are presented as cases per 100,000 population rather than percentages. This approach is consistently adopted across GBD publications for two key reasons:

Enhanced Comparability: Rates per 100,000 facilitate direct comparisons across regions and time periods, especially for conditions with low prevalence (e.g., pediatric near vision loss). Percentages (e.g., 0.1% vs. 0.2%) may obscure meaningful differences in disease burden when absolute numbers are small.

Alignment with GBD Methodology: The GBD Study universally reports prevalence, incidence, and disability metrics as rates per 100,000 to maintain consistency across diverse diseases and populations. This convention ensures harmonization with prior GBD-based research and minimizes interpretive ambiguity.

In response to the issues you raised, we have added annotations to the y-axis of the figures to facilitate readers' comprehension and understanding. Meanwhile, we have now explicitly stated the rationale for using rates per 100,000 in the Methods section (Line 138-140).

Comment 5: The manuscript uses many abbreviations, which can hinder readability. It is recommended to limit abbreviations to widely recognized terms or include a table listing all abbreviations and their meanings to help readers follow the content more easily.

Response: Your suggestions are highly constructive. After troubleshooting, we have retained only the full name of the term as it first appeared, replacing it only with shorthand in the subsequent text. In addition, we have created a new glossary containing abbreviations and simple definitions of terms to improve readability, which can be seen in Table 3.

Comment 6: Some figures lack units on the axes. For example, graphs reporting prevalence cases do not specify units, which makes the data presentation unclear. Additionally, presenting prevalence over time with only case numbers can be misleading and should be corrected.

Response: Thank you very much for your meticulous observation. As we mentioned in Response 4, the presentation of the rate is in terms of the number of individuals per 100,000 people. Your suggestion is very meaningful, and we have now supplemented the labeling on the y-axis of the figures to make it more standardized.

Reviewer #3: In this manuscript the authors have presented a clear picture of the global burden of near vision loss and its future projections in the age group of under 20 years.

Comment 1: The authors use “australasia” in the abstract and tables. I am unaware of such a regional term. Please define the term or edit it to a more familiar term.

Response: Thank you very much for your suggestions. In the GBD study, ‘Australasia’ refers to a region of Oceania that includes Australia, New Zealand and neighbouring Pacific islands, which is a high-income region.

Comment 2: In the abstract “Notably, NVL burden in higher SDI regions rebounded post-COVID-19, reversing previ

---

## [Decision Letter · Decision Letter 1]

Global, regional, and national burden of near vision loss in children and adolescents under 20 years from 1990–2021 and prediction to 2060: a cross-sectional study based on the Global Burden of Disease Study 2021.

PONE-D-25-07176R1

Dear Dr. Zhou,

We’re pleased to inform you that your manuscript has been judged scientifically suitable for publication and will be formally accepted for publication once it meets all outstanding technical requirements.

Kind regards,

Ali Faramarzi, MD, MPH

Academic Editor

PLOS ONE

Additional Editor Comments (optional):

Reviewers' comments:

Reviewer's Responses to Questions

**Comments to the Author**

Reviewer #1: All comments have been addressed

Reviewer #2: All comments have been addressed

Reviewer #3: All comments have been addressed

2. Is the manuscript technically sound, and do the data support the conclusions?

Reviewer #1: Yes

Reviewer #2: Yes

Reviewer #3: Yes

3. Has the statistical analysis been performed appropriately and rigorously?

Reviewer #1: Yes

Reviewer #2: N/A

Reviewer #3: Yes

4. Have the authors made all data underlying the findings in their manuscript fully available?

Reviewer #1: Yes

Reviewer #2: Yes

Reviewer #3: Yes

5. Is the manuscript presented in an intelligible fashion and written in standard English?

Reviewer #1: Yes

Reviewer #2: Yes

Reviewer #3: Yes

Reviewer #1: (No Response)

Reviewer #2: Thank you for being receptive to feedback and addressed all my comments and reflect it in the revision.

Reviewer #3: (No Response)

**Do you want your identity to be public for this peer review?** For information about this choice, including consent withdrawal, please see our Privacy Policy

Reviewer #1: No

Reviewer #2: **Yes: ** Anh V Bui

Reviewer #3: No

---

## [Editor Report · Acceptance letter]

PONE-D-25-07176R1

PLOS ONE

Dear Dr. Zhou,

I'm pleased to inform you that your manuscript has been deemed suitable for publication in PLOS ONE. Congratulations! Your manuscript is now being handed over to our production team.

Kind regards,

on behalf of

Dr. Ali Faramarzi

Academic Editor

PLOS ONE